# Prevalence and pattern of rheumatic valvular heart disease in Africa: Systematic review and meta-analysis, 2015–2023, population based studies

**Seid Mohammed Abdu** [ID]*, **Altaseb Beyene Kassaw, Amare Abera Tareke, Gosa Mankelkl, Mekonnen Belete** [ID]**, Mohammed Derso Bihonegn, Ahmed Juhar Temam, Gashaw Abebe, Ebrahim Msaye Assefa** [ID]

Department of Biomedical Sciences, College of Medicine and Health Sciences, Wollo University, Dessie, Ethiopia

* seidmd041@gmail.com

## Abstract

### Background

Rheumatic heart disease is a global health concern with a persistently high incidence in developing countries, including Africa. It has a significant economic, morbidity, and mortality burden, particularly for children and young adults during their most productive years. However, in the last ten years, the extent of its impact in Africa has remained unclear. Limited studies conducted on the continent have reported diverse prevalence rates of rheumatic heart disease. As a result of these, the study aimed to aggregate and synthesize findings from population-based studies to offer a comprehensive and updated overview of rheumatic heart disease prevalence and pattern at the African level.

### Methods

The studies were identified through a comprehensive literature search of the electronic databases, including PubMed, Google Scholar, Web searches, and manual searches. The descriptive information for the study is presented in the table, and the quantitative results are presented in forest plots. The Cochrane Q test and $I^2$ test statistic were used to test heterogeneity across studies. The pooled estimate of the prevalence of rheumatic heart disease was computed by a random effects model.

### Results

Out of 22 population-based studies analyzed using random-effects, the pooled magnitude of rheumatic heart disease was found to be 18.41/1000 (95% CI: 14.08–22.73/1000). This comprised definite cases of rheumatic heart disease at a prevalence rate of 8.91/1000 (95% CI: 6.50–11.33/1000) and borderline cases at a prevalence rate of 10.69/1000 (95% CI: 7.74–13.65/1000). The combined prevalence of rheumatic heart disease in males was almost equivalent to that in females. Mitral valve regurgitation was the predominant valve affected by rheumatic heart disease, accounting for approximately 73%.

**Data Availability Statement:** All relevant data are within the paper and its Supporting Information files.

**Funding:** The author(s) received no specific funding for this work.

**Competing interests:** The authors have declared that no competing interests exist.

**Abbreviations:** RHD, Rheumatic Heart Disease; RHVD, Rheumatic valvular Heart Disease; MeSH, Medical Subject Heading; WHF, World heart federation.

## Conclusion

This study analysis found the prevalence of rheumatic heart disease in Africa is high. Because of this, policies and interventions should give attention to prioritize continuous population based active surveillance for early detection of cases to the reduction of rheumatic heart disease sequel, especially in the children and adolescent population.

## Introduction

Rheumatic Valvular Heart Disease (RVHD) is an abnormal post-infectious sequel of acute rheumatic fever (ARF) resulting from a harmful immune response to a streptococcal pharyngitis that causes valvular damage [1]. The disease of the heart valves is caused by one or several episodes of rheumatic fever and an autoimmune inflammatory reaction to a throat infection caused by group A streptococci [2].

Around the world, 30 million people are currently believed to be affected by rheumatic heart disease (RHD) [3]. From this global burden in 2015, 84% of all available cases and 80% of the estimated deaths were from African, Southeast Asian, and Western Pacific regions [3]. RVHD typically occurs during childhood and has the potential to result in either death or life-long disability [2]. Globally, in 2015, RHD was estimated to have caused 305 000 deaths and 11.5 million disabilities [4]. In areas where RHD is prevalent, it stands out as the primary heart ailment in adolescent girls and pregnant women, leading to substantial maternal and perinatal morbidity and mortality [4].

New cases of RHD are influenced by multiple factors such as overcrowding, inadequate housing, malnutrition, limited access to healthcare, family history, age, pregnancy, and gender [5–8]. Countries with a persistently high incidence of RHD face a significant economic burden, particularly impacting children and young adults during their most productive years. Because of RHD, worldwide in 2010, it cost up to 5.4 trillion USD [4]. To tackle these negative impacts of RHD, the African Union Heads of State and Government [9], along with the Global Action Plan for Non-communicable Diseases and the United Nations [10], acknowledge the urgency of addressing RHD to reduce early death and disability. They emphasize the importance of early screening, intervention, control, and elimination of RHD to achieve global health goals [10].

To the best of our knowledge, there has been no recent systematic assessment of RVHD in Africa. Therefore, conducting an examination of the pooled magnitude of RVHD among the general population is crucial for informed decision-making and the formulation of preventive and early treatment strategies. As a result, our research question focuses on determining the overall magnitude and pattern of RVHD in Africa. Because of the diverse sociodemographic landscape, limited healthcare resources, and financial constraints in Africa, managing RVHD and its associated complications is a bit complex. This information, therefore, is essential for gaining a scientific understanding of the burden posed by RVHD on the population and can serve as a baseline for the formulation of preventive measures on the continent.

## Materials and methods protocol registration

This systematic review and meta-analysis was conducted to synthesize existing evidence on the prevalence of RHD in Africa. The protocol was registered by the International Prospective Register of Systematic Reviews (PROSPERO) with registration number [CRD42024513071].

## Search strategy

This systematic review and meta-analysis was carried out according to the Preferred Reporting Items for Systematic Reviews and Meta-Analyses (PRISMA) guidelines [11] (**S1 Checklist**). A comprehensive literature search of the electronic databases, including PubMed, Google Scholar, Web search, and Hinari, was carried out up to January 31, 2024. The search strategy was performed in three stages. In the first stage, relevant Medical Subject Headings (MeSH) and other terms were identified in the literature. In the second phase, full searches were conducted in the mentioned databases. In the third phase, the bibliographies of relevant studies and university websites were searched to see the presence of eligible studies. The databases were searched by combining different MeSH terms and keywords using Boolean Operator (OR, AND). For example, for PubMed, the following search strategy was entered: ((((((Prevalence*[Title/Abstract])OR(Magnitude*[Title/Abstract])) OR(Incidence*[Title/Abstract])) OR (Proportion*[Title/Abstract])) OR (Epidemiology[Title/Abstract])) AND ((((((Disease, Rheumatic Heart[Title/Abstract]) OR (Diseases, Rheumatic Heart[Title/Abstract])) OR (Heart Disease, Rheumatic[Title/Abstract])) OR (Heart Diseases, Rheumatic[Title/Abstract])) OR (Rheumatic Heart Diseases[Title/Abstract])) OR (Rheumatic Heart Disease[Title/Abstract]) AND ((y_10[Filter]) AND [Countries name]. filters; English, free full text, and date from 2015- January 31, 2024, (**S1 Appendix**).

Criteria for considering studies for the review.

## Inclusion criteria

**Design.** Population-based observational studies.

**Population.** Any age group in the population participated in the study.

**Settings.** Only population-based studies conducted in African nations.

**Language.** The only articles considered in this review were those written in English. Publication or report year: The study will include the past 9 years for systematic review and meta-analysis.

**Method of diagnosis.** All studies used echocardiography of diagnostic criteria, with the World Heart Federation (WHF), and WHO to diagnose rheumatic valvular heart disease.

**Outcome.** Magnitude of RHD and its pattern.

## Exclusion criteria

The following research is excluded from selection: Institutional-based studies, case report and case series studies, as well as any other studies that lack the pertinent information required to estimate the magnitude of RVHD.

## Data extraction and quality assessment

Two investigators (SMA and EMA) independently assessed the eligibility of articles, and disagreements were solved by objective measurement. The same two authors independently extracted data using Microsoft Excel. The data extraction sheet included author name, year of publication, country, valve involvement, sample size, and number of RHD patients. Any discrepancies between investigators were resolved by consensus. Among the articles identified, titles and abstracts were reviewed to retrieve studies on the magnitude of RVHD. Articles found relevant by title and abstract were screened for full eligibility. The methodological quality of the included studies was assessed using the quality assessment tool of the JBI for prevalence studies [12]. Good-quality articles were determined if the scale score was 5 or above [12] (**S2 Appendix**).

## Outcome of interest

The primary outcome of this study was the magnitude of RHD and pattern, which was reported in the original paper as a percentage and/or as the number of RHD cases (n) per total number of assed patients (N).

## Statistical analysis

The pooled magnitude of RHD, accompanied by a corresponding 95% CI, was calculated using the random-effects, inverse variance method. Sensitivity analysis, publication bias and heterogeneity were also assessed. The heterogeneity of studies was checked using Cochran's Q test and $I^2$ test statistics. A p-value of $< 0.05$ for the $I^2$ test was used to determine the presence of heterogeneity. Potential publication bias and small study effects were assessed using funnel plots, and where possible, Egger's regression test was performed. The p-value $< 0.05$ cut-point was used to declare statistical significance. To explore potential sources of heterogeneity, a pre-defined subgroup analysis was performed based on the region, year, sampling unit, and countries where the studies were conducted and a P-value of $<0.05$ was considered significant. Additionally, sensitivity analysis was carried out using a leave-one-out method to look into the impact of individual studies and the stability or robustness of the pooled values to outliers. Statistical analyses were done via STATA version 17 (STATA Corporation, College Station, TX, USA).

## Results

We identified a total of 1831 publications in our initial search; out of one thousand eight hundred thirty one identified studies, 18 articles detailing 22 population-based studies fulfilled the eligibility criteria and were incorporated into the systematic review and meta-analysis. Notably, six of these articles were included from two distinct population-based studies [13,14]. The supplementary materials provide an overview of the methodological characteristics of the included studies. In 15 (68.2%) of the population-based studies, schools served as the primary sampling units, while in the remaining 7 (31.8%), communities were the source of sampling units. In our study, echocardiography assessment was the primary screening method, and WHF was used as the criteria for detection of RHD among different studies (**Fig 1**).

## Characteristic of studies

The meta-analysis included a total of 22 population-based studies with a total population of 45071. Around two-thirds (13, 59%) of the studies were conducted in East Africa, followed by West Africa (5, 22.72%), South Africa (4, 18.18%), and finally one study was included from North Africa. The sample size of the studies ranged from 417 to 4572. The highest (61.54/1000) and the lowest (2.33/1000) prevalence of RVHD were reported in studies conducted in Sudan. Four and five studies were conducted in Sudan [15–17] and Ethiopia [13,18–20], respectively (**Table 1**).

## The pooled magnitude of rheumatic valvular heart disease in Africa

The pooled magnitude of RVHD in Africa was 18.41/1000 (95% CI: 14.08/1000 to 22.73/1000), and the inverse variance $I^2$ was 96% with P<0.001, indicating heterogeneity in the reported prevalence of RVHD among the included studies (**Fig 2**). This heterogeneity could be a result of differences in study setting, study region, or screening criteria for RHD used by the included studies. This study also disclosed the prevalence of RVHD among males and females. Fifteen

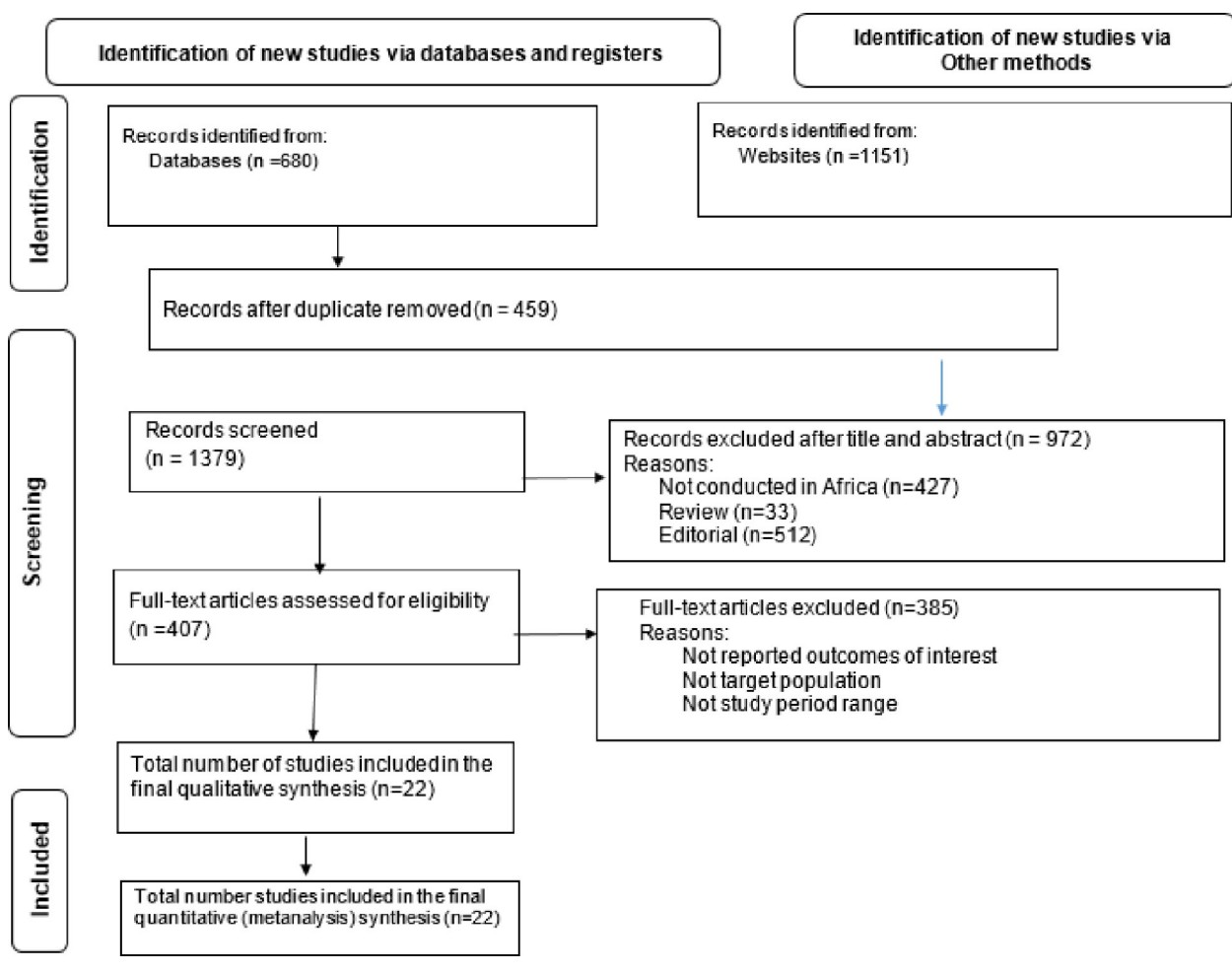

**Fig 1. Depicts the schematic flow of study selections steps.**

studies provided data on the prevalence of RVHD in both genders. The combined prevalence of RVHD in males is almost equivalent to that in females, with rates of 10.15 per 1000 (95% CI: 6.84, 13.47) and 9.72 per 1000 (95% CI: 6.46, 12.98), respectively (**Table 1**). The study also indicated that among cases of RVHD, mitral valve regurgitation is the predominant valve affected, accounting for approximately 73% of the total identified cases across 14 included studies (**Fig 3**). Aortic valve regurgitation, on the other hand, constituted 7% of the cases. Simultaneous involvement of both the aortic and mitral valves in regurgitation was observed in around 8% of cases. The study furthermore revealed that the pooled prevalence of definite RHD (8.91 per 1000, 95% CI 6.50 to 11.33) was about slightly lower than that of borderline disease (10.69 per 1000, 95% CI 7.74 to 13.65).

## Sensitivity analysis

A leave-one-out sensitivity analysis was carried out to see whether the findings of a single study had a significant impact on the pooled prevalence of RHD in Africa. However, all the results of this sensitivity analysis were within the 95% CI limits of the pooled prevalence (14.08 to 22.73/1000), indicating that no significant study may have had an impact on the observed pooled prevalence of RHD (**Fig 4**).

Table 1. Description of studies included in the meta-analysis of the prevalence and pattern of rheumatic heart disease in Africa.

| First Author name, Year | Country | Sampling unit | Mean Age | RR% | Sample size Included | Prevalence per 1000 | | | Male-RHD* | female-RHD | Prevalence per 1000 of RHD with CI **among | |
|---|---|---|---|---|---|---|---|---|---|---|---|---|
| | | | | | | RHD with 95% CI | Definite | Borderline | | | Male | Female |
| Tadesse Gemechu et al (2017)[19] | Ethiopia | Community | 13 | 82 | 987 | 56.74 (43.95,72.96) | 37.49 | 19.25 | 28 | 28 | 28.37 (19.70,40.70) | 28.37 (19.70, 40.70) |
| Sulafa Ali et al (2017)[21] | Sudan | Community | 15 | 100 | 3315 | 61.54 (53.85,70.24) | 40.12 | 21.42 | 96 | 108 | 28.96 (23.77,35.24) | 32.58(27.06, 39.18) |
| Amy Scheel et al (2018)[22] | Uganda | Community | 20 | 83.6 | 2453 | 24.46 (19.05,31.36) | 12.64 | 11.82 | 24 | 36 | 9.78 (6.58, 14.52) | 14.68 (10.62, 20.25) |
| Mark E Engel et al (2015)[23] | Ethiopia | School | 10.7 | 99 | 2000 | 30.50 (23.82,38.98) | 16.50 | 14.00 | – | – | – | – |
| Mark E Engel et al (2015)[23] | S. Africa | School | 12.2 | 94.3 | 2720 | 20.22 (15.57,26.23) | 4.78 | 15.44 | – | – | – | – |
| Mark E Engel et al (2015)[23] | S. Africa | School | 11.3 | – | 1279 | 12.51 (7.715,20.22) | 2.35 | 10.16 | – | – | – | – |
| Mark E Engel et al (2015)[23] | S. Africa | School | 13.1 | – | 1441 | 27.10 (19.86,36.78) | 6.94 | 20.13 | – | – | – | – |
| Ahmed Ali etal (2023)[24] | Egypt | School | 13.1 | 92.86 | 1560 | 23.10 (16.72,31.78) | 23.08 | – | 24 | 12 | 15.39 (10.36,22.79) | 7.69 (4.41,13.40) |
| Ekanem N. Ekure( 2019)[25] | Nigeria | School | 11.3 | 100 | 4107 | 2.68 (1.496, 4.79) | 0.49 | 2.19 | 7 | 4 | 1.70 (0.83,3.51) | 0.97 (0.38,2.50) |
| Aliou Alassane et al(2015)[26] | Senegal | School | 9.7 | 100 | 2019 | 4.95(2.69, 9.09) | 4.95 | – | 6 | 4 | 2.97 (1.36, 6.47) | 1.98 (0.77,5.08) |
| Dr.Hailu Abera et al(2016)[20] | Ethiopia | School | 8.86 | 100 | 1874 | 3.20(1.47 6.97) | 3.20 | – | 2 | 4 | 1.07 (0.29,3.88) | 2.13 (0.83,5.48) |
| Dejuma Yadeta et al (2016)[18] | Ethiopia | School | 13.22 | 98.1 | 3238 | 18.22 (14.15, 23.43) | 13.59 | 4.63 | 18 | 26 | 5.56 (3.52, 8.77) | 8.03(5.49, 11.74) |
| J. mucumisti et al(2017)[27] | Rwanda | School | 11.2 | 83.3 | 2501 | 6.80 (4.25 10.86) | 1.60 | 5.20 | 13 | 4 | 5.20 (3.04,8.87) | 1.60(0.62, 4.11) |
| John Musuku et al(2018)[28] | Zambia | School | 15.4 | 100 | 1102 | 11.80(6.91, 20.10) | 2.72 | 9.07 | 5 | 8 | 4.54 (1.94,10.58) | 7.26 (3.68,14.26) |
| Amy Sims Sanyahumbi etal(2016)[29] | Malawi | School | – | 100 | 1450 | 33.79(25.66, 44.39) | 6.90 | 26.90 | 20 | 29 | 13.79 (8.95, 21.21) | 20 (13.96,28.58) |
| Parvina Titus Kazahura et al (2021)[30] | Tanzania | Community | 10.8 | 100 | 949 | 33.72(23.99, 47.21) | 17.91 | 15.81 | 23 | 9 | 24.24 (16.20,36.11) | 9.48 (5.0,17.93) |
| Sulafa Ali et al (2018)[15] | Sudan | Community | 10.5 | 100 | 3000 | 2.33(1.13, 4.81) | 0.33 | 2.00 | – | – | 6.01 (3.16,11.38) | 13.35 (8.66, 20.53) |
| Sulafa Ali et al (2018)[15] | Sudan | Community | 10.8 | 98.9 | 1498 | 19.36 (13.51, 27.67) | 14.69 | 4.67 | 9 | 20 | – | – |
| Sulafa Ali et al (2018)[16] | Sudan | Community | – | 92 | 2129 | 2.35(1.00, 5.49) | 1.88 | 0.47 | – | – | – | – |

*(Continued)*

**Table 1.** (Continued)

| First Author name, Year | Country | Sampling unit | Mean Age | RR% | Sample size Included | Prevalence per 1000 | | | Male-RHD* | female-RHD | Prevalence per 1000 of RHD with CI **among | |
|---|---|---|---|---|---|---|---|---|---|---|---|---|
| | | | | | | RHD with 95% CI | Definite | Borderline | | | Male | Female |
| Sulafa Ali et al (2022)[17] | Sudan | Community | – | 100 | 4572 | 10.28(7.74, 13.64) | – | – | – | – | – | – |
| Esin Nkereuwem et al(2020)[31] | Nigeria | School | – | 100 | 417 | 21.58(11.40, 40.50) | 2.40 | 19.19 | 7 | 2 | 16.79 (8.16,34.24) | 4.80(1.32, 17.32) |
| Ujuanbi A. et al (2019)[32] | Nigeria | School | 10.29 | 100 | 461 | 6.51(,2.22 18.96) | 6.51 | – | – | 3 | – | 6.51 (2.22,18.96) |
| Pooled prevalence | – | – | – | – | – | 18.41(14.08 to 22.73) | 8.91 (6.50,11.3) | 10.69 (7.74 to 13.65) | – | – | 10.16 (6.84 to 13.47) | 9.72(6.46 to 12.98) |

*Rheumatic Heart Disease

** Confidence interval.

## Publication bias

Due to the asymmetrical distribution observed in the funnel plot (**Fig 5**), we proceeded to perform Egger's regression test. The test yielded a significant result, with a p-value less than 0.001, suggesting the presence of publication bias in the included studies.

To reduce and adjust the publication bias in the studies, non-parametric trim and fill analysis was performed for estimation of the number of missing studies that might exist. During this analysis, three studies were imputed, and after adjustment of publication bias, the estimated pooled prevalence of RHD was 21.96 per1000. 95% CI (14.56–29.34) (**Fig 6**).

## Subgroup analysis of rheumatic heart disease

A significant heterogeneity was seen in the primary studies that were incorporated into this systematic review and meta-analysis. So as to look into the sources of heterogeneity, a subgroup analysis based on the region where the studies were conducted was carried out. The subgroup analysis revealed the source of heterogeneity in the studies carried out in East Africa ($I^2$ = 97.2%, p<0.001), West Africa ($I^2$ = 91.6%, p<0.001), and South Africa ($I^2$ = 73.2%, p = 0.011) (**Fig 7**).

Subgroup Analysis of RHD further was carried out by study Countries and sampling unit, the pooled point estimate prevalence of rheumatic heart disease in Ethiopia was 25.83/1000 (95% CI: 9.56 to 42.1/1000) in Sudan, it was 18.23/1000 (95% CI: 7.83 to 28.63/1000), in South Africa, 19.53/1000 (95% CI: 12.03 to 27.04) and in Nigeria, it was 7.60/1000 (95% CI: 0.66 to 15.85/1000) and based on study area, conducted in community ($I^2$ = 98 p< 0.0001 and in school ($I^2$ = 93.6, P<0.0001)

## Discussion

RHD represents a preventable yet significant public health concern in low- and middle-income nations, including Africa [33]. This is particularly evident in endemic regions characterized by high levels of poverty, crowded living conditions, inadequate housing, malnutrition,

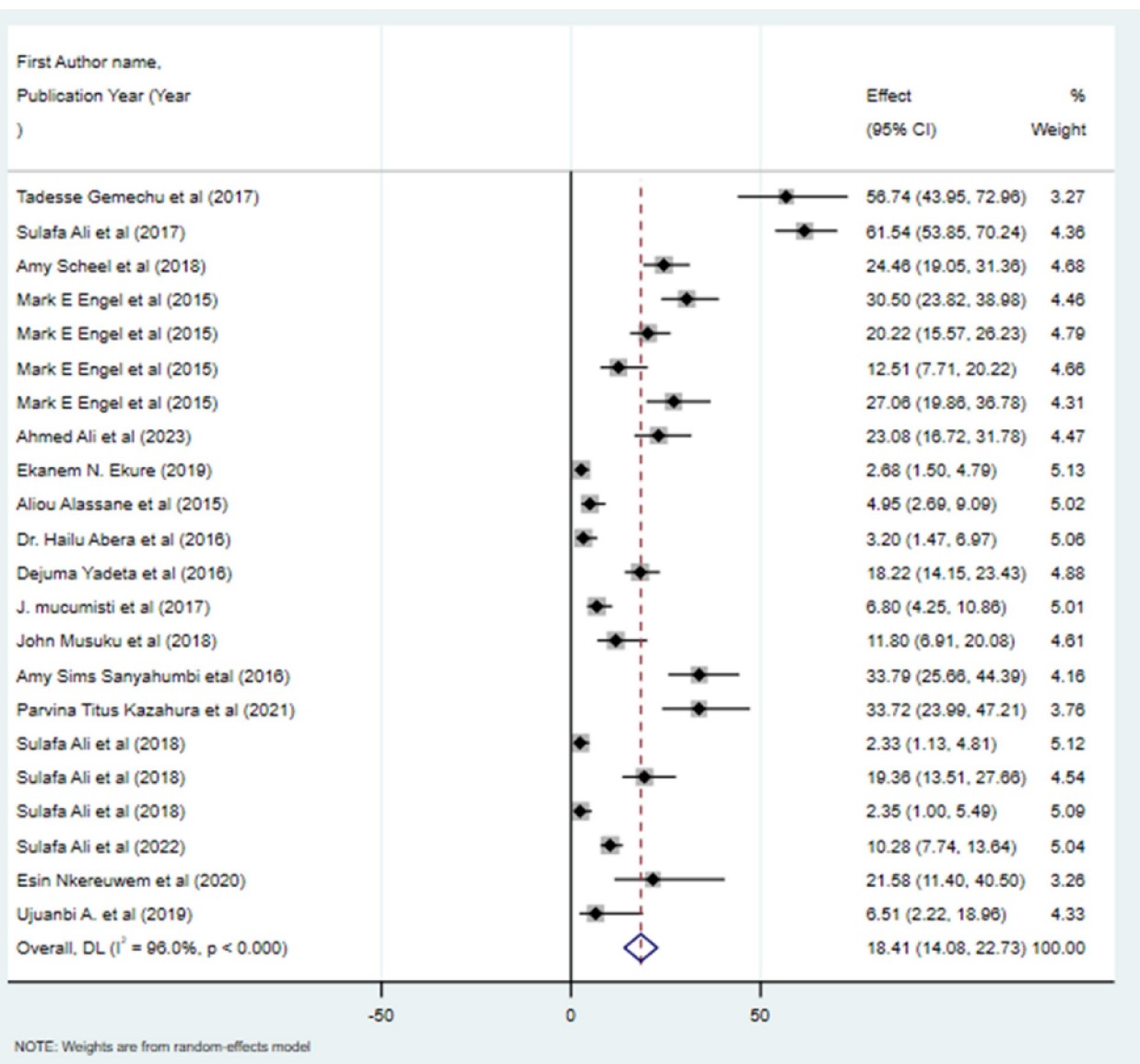

**Fig 2. Forest plot depicting the pooled magnitude of RHD in Africa.**

and limited access to healthcare services [34,35]. Consequently, this meta-analysis aimed to update the pooled magnitude of RHD in Africa for the formulation of preventive measures.

The studies included in our analysis utilized echocardiography for screening, following the criteria established by WHF to ensure standardized criteria for epidemiological screening and international echocardiographic standardization [36,37]. Other countries conducting similar studies using echocardiography have found it to be much more sensitive, especially for active surveillance. For instance, in a school-based screening study from Cambodia and Mozambique [38], showed nearly ten times more cases of RHD when using echocardiography compared to clinical evaluation alone. Fiji [39], New Zealand [40], and a meta-analysis involving children and adolescents [41] also reported high rates of RHD through echocardiographic screening.

The final analysis of our study resulted comprehensive pooled prevalence of 18.41/1000 (95% CI: 14.08–22.73). We noticed a significant prevalence of RHD, accompanied by

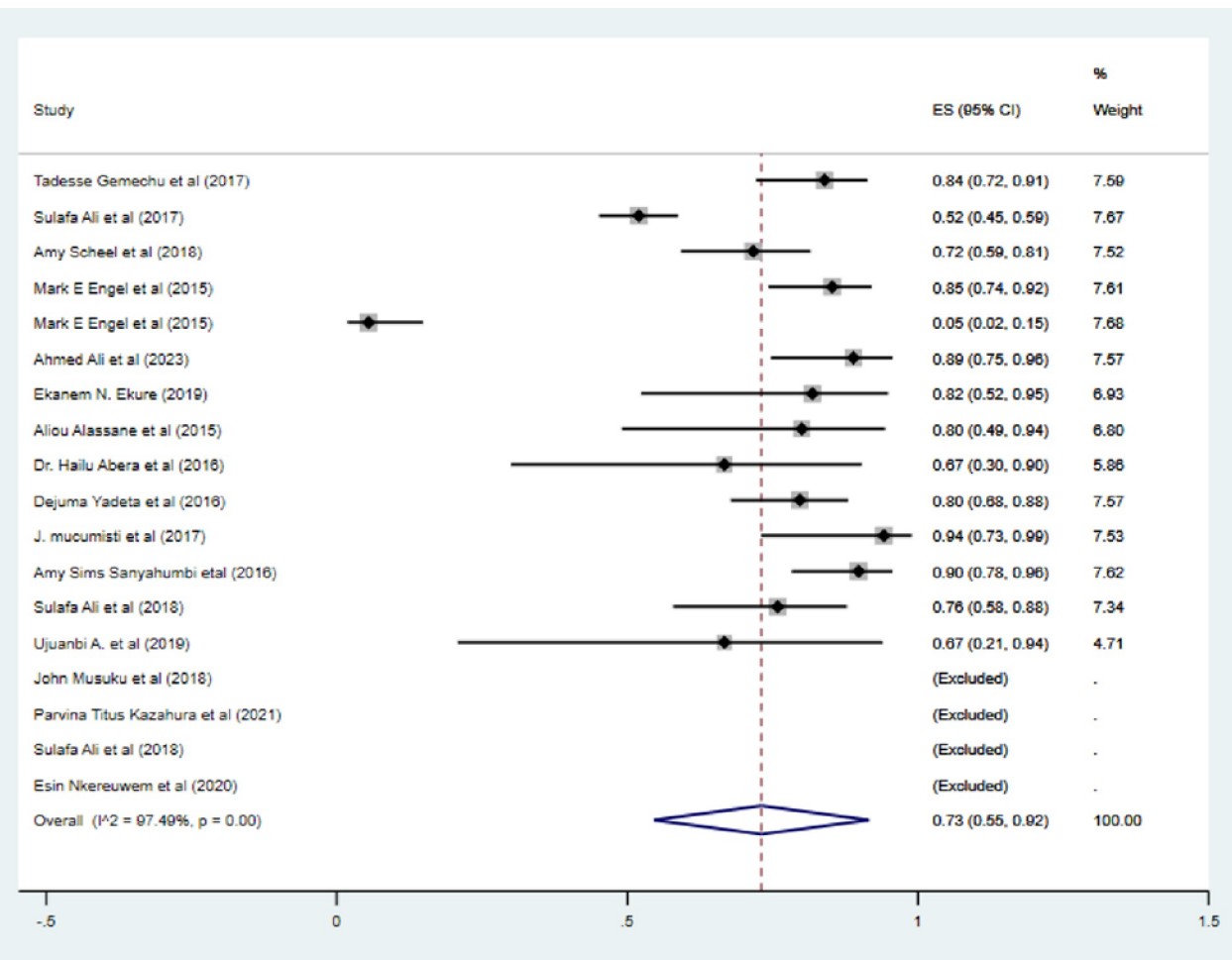

**Fig 3. Forest plot depicting the pooled magnitude of mitral valve regurgitation in Africa.**

significant heterogeneity in the finding. It is comparable to a study conducted in South Asia, with a magnitude of 18.28/1000 [42]. On the contrary, a global systematic review and meta-analysis of population-based echocardiographic studies revealed a higher prevalence of RHD, reaching 26.1% [43]. Additionally, East Africa's meta-analysis showed a prevalence of 14.67% [44]. The difference in these results may be because our analysis focused on data from 2015–2024, while East Africa's study included all publications until December 2019, and East Africa's study utilized multiple diagnostic methods endorsed by WHO or WHF, including echocardiography, clinical auscultation, ECGs, and radiography. In contrast, our approach exclusively relied on WHF criteria and echocardiography. Likewise, the East Africa meta-analysis covered both community and health institution studies, but our study specifically concentrated on community-based research only.

In our analysis, borderline RHD slightly exceeded the prevalence of definite RHD. Borderline RHD is a diagnostic category designed by WHF, which prioritizes sensitivity over specificity [45]. Despite this emphasis on sensitivity, meeting borderline RHD criteria doesn't always indicate the presence of a diseased state [36]. However, in the African context, the significant occurrence of borderline RHD should not be dismissed because borderline RHD has a stronger correlation with high-risk populations compared to low-risk RHD [46]. For example, 16%

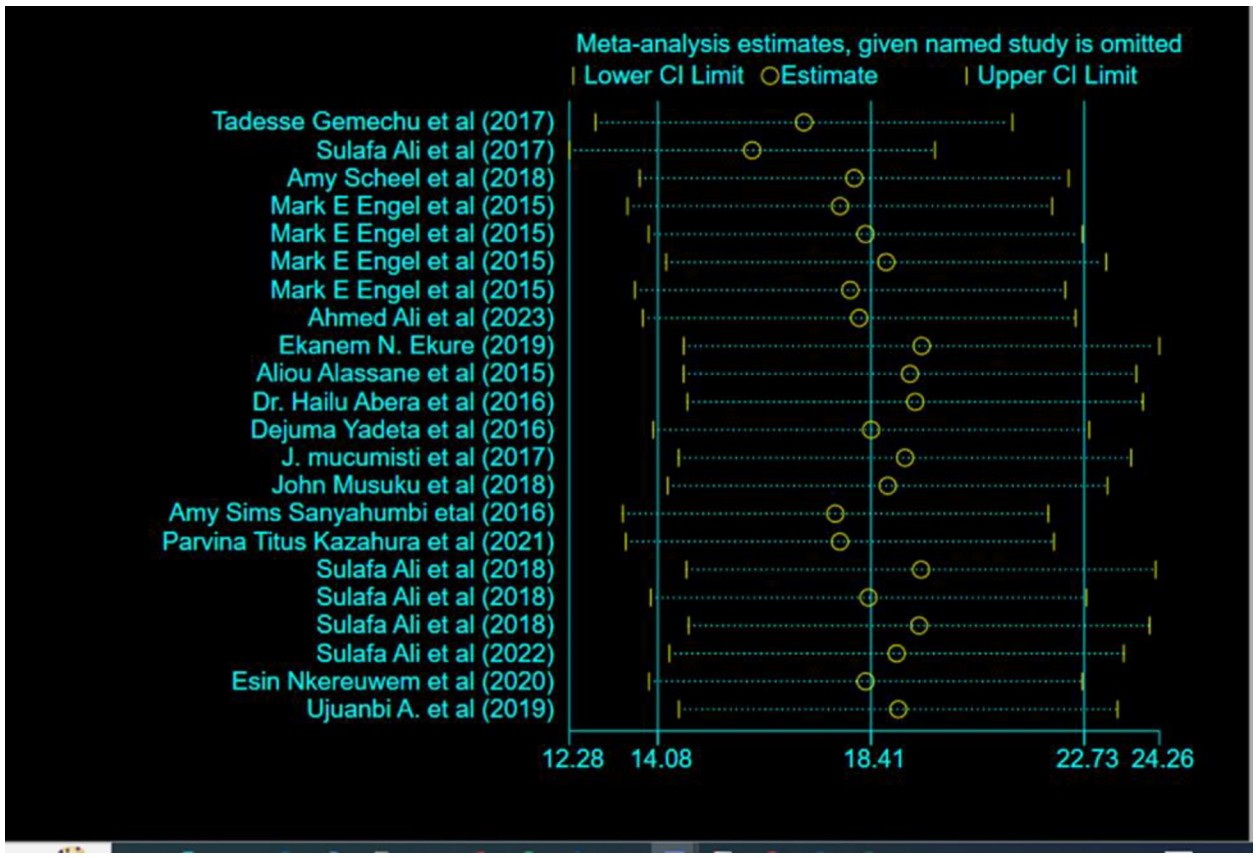

**Fig 4. Sensitivity analysis of magnitude for RHD in Africa.**

of individuals in South Africa [47] and 11% of individuals in Nicaragua [48] initially diagnosed with borderline RHD had progressed to definite RHD within two to five years of follow-up. This highlights the need for ongoing follow-up among individuals with borderline RHD. Moreover, the notable presence of borderline RHD, especially without a history of rheumatic fever, indicates a hidden disease that can only be found through regular echocardiographic screening. Therefore, implementing timely diagnostic strategies for borderline rheumatic heart disease can help to prevent or slow the progression of valvular damage [49].

Females are at a greater risk of developing RHD compared to males due to higher exposure to β-haemolytic streptococci over time [50] and hormonal effects on the immune system [51]. This evidence is supported by studies conducted in different regions of the world [52–54]. However, our analysis did not find any significant difference in RHD burden between sexes. Even if it's challenging to pinpoint the exact reasons why the burden of RHD is nearly equal between sexes for our study, the possible reason might be that our analysis utilized studies that used standard diagnostic criteria and more sensitive screening methods, which could result in a more accurate estimation of RHD prevalence across genders.

Among the 14 studies included in our analysis, the prevalence of mitral regurgitation was 73%; both aortic and mitral valve regurgitation were observed in around 8% of cases, followed by 7% with aortic regurgitation. These observations confirm the high prevalence of mitral valve involvement in RHD. So, it's important to carefully check the heart valve to find people with mitral valve regurgitation, especially in areas where RHD is common and there's a history of rheumatic fever.

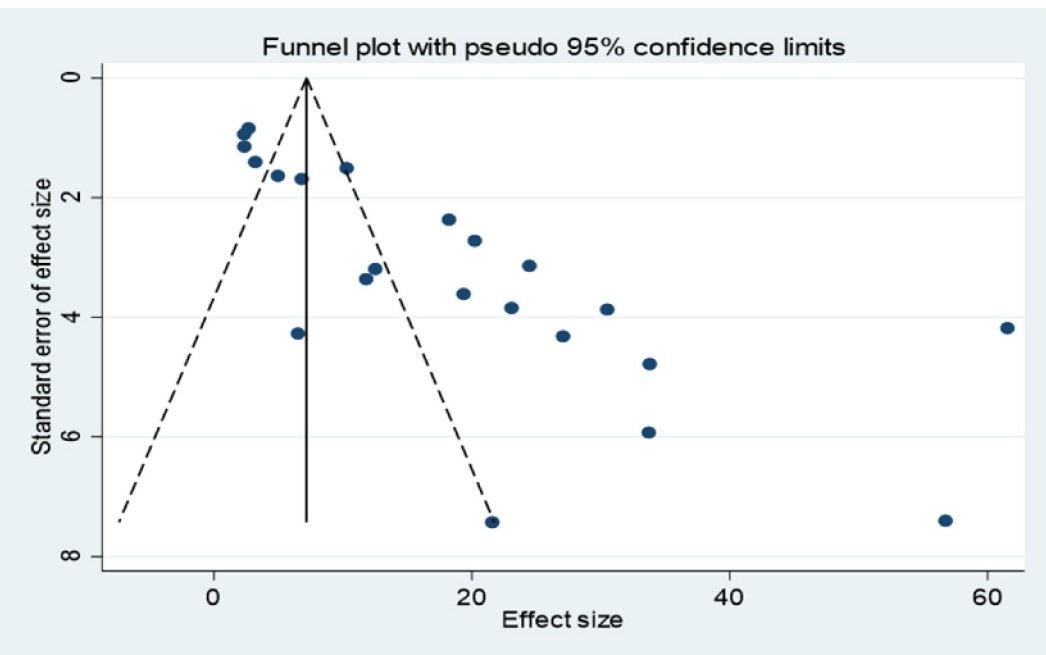

**Fig 5. Funnel plot of the risk of publication bias for the magnitude of RHD, in Africa.**

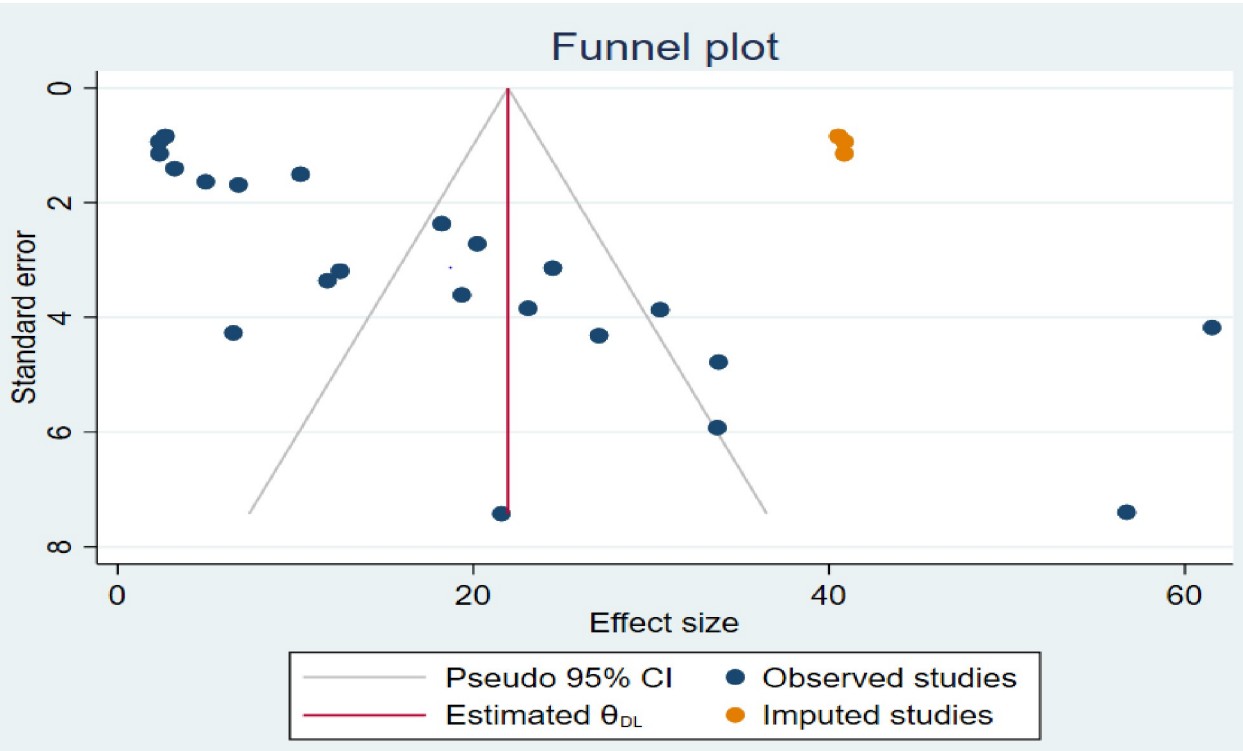

**Fig 6. Funnel plot of non-parametric trim and fill analysis of publication bias for the magnitude of RHD, in Africa.**

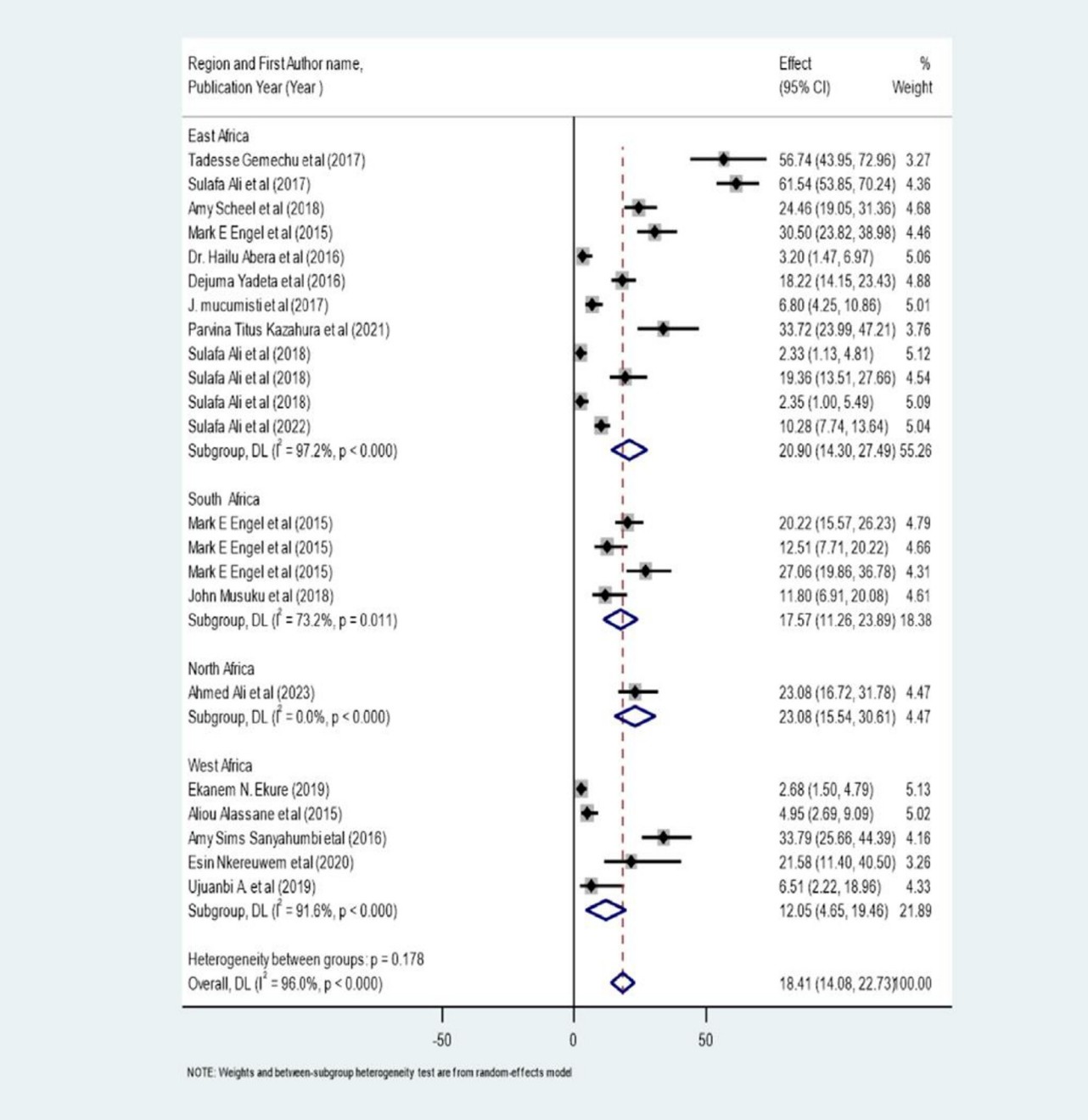

**Fig 7. Forest plot showing the subgroup analysis by region for the pooled magnitude of RHD in Africa.**

Our review primarily relies on samples derived from school lists, which may underestimate the actual prevalence of disease. This is because school attendance is strongly linked to socio-economic status, a significant risk factor for RHD [35,55]. The response rates were varied among the included studies; this may induce bias, which makes it difficult to generalize for Africa. Our review is also susceptible to considerable heterogeneity among studies, potentially influencing the outcomes of the meta-analysis. Finally, an additional limitation we encountered in our review was the potential presence of publication bias. To mitigate this bias, we conducted a non-parametric trim and fill analysis. Through this method, we estimated the

number of missing studies and imputed three additional studies. Following the adjustment for publication bias, the estimated pooled prevalence of RHD was determined to be 21.96/1000 (95% CI 14.56–29.34).

## Limitation of the study

Our inclusion criteria were limited solely to English-language studies, as it could result in the omission of valuable study data in languages other than English. This limitation has the potential to impact the comprehensiveness and generalizability of our findings.

## Conclusions

The analysis of RHD prevalence in Africa revealed a significant burden, with a pooled magnitude of 18.41/1000. The sex distribution showed a near-equivalence prevalence of RHD. Most cases primarily involved mitral valve regurgitation (73%), followed by both mitral and aortic valves affected in about 8% of cases, with aortic valve regurgitation (7%). Additionally, the prevalence of definite rheumatic heart disease was slightly lower than that of borderline disease. These findings highlight the urgent need for comprehensive strategies for prevention, early detection, and management to mitigate the impact of RHD on public health in Africa.

## Supporting information

**S1 Checklist. PRISMA 2020 checklist.**
(DOCX)

**S1 Table.**
(DOCX)

**S1 Appendix. Search strategies.**
(DOCX)

**S2 Appendix. Quality assessment of included studies.**
(DOCX)

## Author Contributions

**Conceptualization:** Seid Mohammed Abdu.

**Data curation:** Seid Mohammed Abdu, Altaseb Beyene Kassaw, Ebrahim Msaye Assefa.

**Formal analysis:** Seid Mohammed Abdu, Amare Abera Tareke, Ebrahim Msaye Assefa.

**Investigation:** Seid Mohammed Abdu, Gosa Mankelkl, Mekonnen Belete, Mohammed Derso Bihonegn, Ahmed Juhar Temam.

**Methodology:** Seid Mohammed Abdu, Ebrahim Msaye Assefa.

**Software:** Seid Mohammed Abdu, Amare Abera Tareke, Ebrahim Msaye Assefa.

**Supervision:** Seid Mohammed Abdu, Gashaw Abebe, Ebrahim Msaye Assefa.

**Writing – original draft:** Seid Mohammed Abdu, Altaseb Beyene Kassaw, Gosa Mankelkl, Mekonnen Belete, Mohammed Derso Bihonegn, Ahmed Juhar Temam, Gashaw Abebe, Ebrahim Msaye Assefa.

**Writing – review & editing:** Seid Mohammed Abdu, Altaseb Beyene Kassaw, Amare Abera Tareke, Gosa Mankelkl, Mekonnen Belete, Mohammed Derso Bihonegn, Ahmed Juhar Temam, Gashaw Abebe, Ebrahim Msaye Assefa.

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
