## [Editor Report · Decision Letter 0]

22 Mar 2024

PONE-D-24-09272Prevalence and Pattern of Rheumatic valvular Heart Disease in Africa Systematic Review and Meta-Analysis, 2015-2023, Population-Based StudiesPLOS ONE

Dear Dr. Abdu,

Thank you for submitting your manuscript to PLOS ONE. After careful consideration, we feel that it has merit but does not fully meet PLOS ONE’s publication criteria as it currently stands. Therefore, we invite you to submit a revised version of the manuscript that addresses the points raised during the review process.

We look forward to receiving your revised manuscript.

Kind regards,

Motunrayo Oluwabukola Adekunle, MBChB, FWACP

Academic Editor

PLOS ONE

Journal Requirements:

-https://apps.who.int/gb/ebwha/pdf_files/WHA71/A71_25-en.pdf

In your revision ensure you cite all your sources (including your own works), and quote or rephrase any duplicated text outside the methods section. Further consideration is dependent on these concerns being addressed.

3. In this instance it seems there may be acceptable restrictions in place that prevent the public sharing of your minimal data. However, in line with our goal of ensuring long-term data availability to all interested researchers, PLOS’ Data Policy states that authors cannot be the sole named individuals responsible for ensuring data access (http://journals.plos.org/plosone/s/data-availability#loc-acceptable-data-sharing-methods).
---

## [Author Response · Author response to Decision Letter 0]

27 Mar 2024

Thank you for your insightful comments and suggestions.

---

## [Editor Report · Decision Letter 1]

10 Apr 2024

Prevalence and Pattern of Rheumatic Valvular Heart Disease in Africa: Systematic Review and Meta-Analysis, 2015-2023, Population-Based Studies

PONE-D-24-09272R1

Dear Dr. Abdu,

We’re pleased to inform you that your manuscript has been judged scientifically suitable for publication and will be formally accepted for publication once it meets all outstanding technical requirements.

Kind regards,

Motunrayo Oluwabukola Adekunle, MBChB, FWACP

Academic Editor

PLOS ONE

Additional Editor Comments (optional):

Dear authors,

Please note the following:

Line 196: when further should be used.

Line 210: remove one school based from the sentence.

Line 232: expunge multiple from the sentence. 3 articles referenced.
---

## [Editor Report · Acceptance letter]

19 Jul 2024

PONE-D-24-09272R1 

PLOS ONE

Dear Dr. Abdu, 

I'm pleased to inform you that your manuscript has been deemed suitable for publication in PLOS ONE. Congratulations! Your manuscript is now being handed over to our production team.

Kind regards, 

on behalf of

Dr. Motunrayo Oluwabukola Adekunle 

Academic Editor

PLOS ONE